# Assessing Surface Texture Features of Asphalt Pavement Based on Three-Dimensional Laser Scanning Technology

**Bo Chen, Chunlong Xiong \*, Weixiong Li, Jiarui He and Xiaoning Zhang**

School of Civil Engineering and Transportation, South China University of Technology, Guangzhou 510006, China; chenb@scut.edu.cn (B.C.); 201810101689@mail.scut.edu.cn (W.L.); 202120108301@mail.scut.edu.cn (J.H.); ctxnzh@scut.edu.cn (X.Z.)
\* Correspondence: cthgclx@mail.scut.edu.cn

**Abstract:** Pavement surface texture features are one of key factors affecting the skid resistance of pavement. In this study, a set of stable and reliable texture measurement equipment was firstly assembled by using the linear laser ranging sensor, control system and data acquisition system. Secondly, the equipment was calibrated, and the superposition error of sensor and control system was tested by making a standard gauge block. Thirdly, four different kinds of asphalt mixture were designed, and their surface texture features were obtained by leveraging a three-dimensional laser scanner. Therefore, the surface texture features were characterized as one-dimensional profile features and three-dimensional surface features. At the end of this study, a multi-scale texture feature characterization method was proposed. Results demonstrate that the measurement accuracy of the laser scanning system in the $x$-axis direction can be controlled ranging from $-0.01$ mm to $0.01$ mm, the resolution in the XY plane is 0.05 mm, and the reconstructed surface model of surface texture features can achieve a good visualization effect. They also show that the root mean square deviation of surface profiles of different asphalt pavements fluctuates greatly, which is mainly affected by the nominal particle size of asphalt mixture and the proportion of coarse aggregate, and the non-uniformity of pavement texture distribution makes it difficult to characterize the roughness of asphalt pavement effectively by a single pavement surface profile. This study proposed a texture section method to describe the 3D distribution of road surface texture at different depths. The macrotexture of the road surface gradually changes from sparse to dense starting from the shallow layer. The actual asphalt pavement texture can be characterized by a simplified combination model of "cone + sphere + column". By calculating the surface area distribution of macro and microtextures of different asphalt pavements, it was concluded that the surface area of asphalt pavement under micro scale is about 1.8–2.2 times of the cutting area, and the surface area of macrotexture is about 1.4 times of the cutting area. Moreover, this study proposed texture distribution density to characterize the roughness of asphalt pavement texture at different scales. The $S_{MA}$ index can represent the macroscopic structure level of different asphalt pavements to a certain extent, and the $S_{MI}$ index can well represent the friction level of different asphalt pavements.

**Keywords:** asphalt pavement; 3D laser scanning technology; texture section method; texture feature; texture distribution density

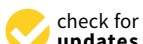

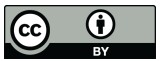

## 1. Introduction

The skid resistance of pavement is an important factor affecting road traffic safety [1]. With the functional design concept of asphalt pavement gradually accepted, the property of the upper layer as the functional wearing course is more prominent. The wearing course is usually designed as a thin layer, which mainly provides skid-resistance, noise reduction and other functions to ensure traffic safety and comfort. In the XVII World Road Congress in Brussels in 1987 by the World Road Association (PIARC), the pavement texture scale is divided into four categories: microtexture, macrotexture, mega texture and

unevenness [2]. The road skid resistance is mainly provided by the macro- and microtexture. The macrotexture determines the drainage capacity of the contact area between the road surface and the tire. The elastic deformation caused by the contact between the tire and the macrotexture will result in hysteresis loss at high speed, which can provide a more stable driving braking force. Furthermore, microtexture mainly provides an adhesion friction component for pavement skid resistance. The sharper the microtexture is, the more easily the pavement texture punctures the water film in the wet condition, which increases the dry contact area between the tire and the pavement and reduces the hydrodynamic lubrication effect between the contact interfaces [3]. The macrotexture depends on the gradation design and construction quality, while the microtexture is related to the mineral composition and crushing process of the aggregate.

At present, the mean texture depth (MTD) is mainly measured by sand patch method, and ASTM E2157 provides a laser cross-section method to measure the mean pavement depth (MPD) [4]. Both of the above-mentioned methods can evaluate pavement macrotexture. The scale of pavement microtexture reaches micron level, so it is too difficult to measure directly, and the pendulum friction coefficient is usually used for indirect evaluation. The sand patch method and the pendulum friction coefficient method are greatly affected by human factors, resulting in large dispersion and low accuracy. Two-dimensional profile measurements can have a relatively high resolution up to the microtexture level, which can evaluate the roughness of an aggregate profile.

In recent years, worldwide researchers have carried out a lot of study on the measurement and evaluation of asphalt pavement texture, which mainly includes contact measurement and non-contact measurement. Contact measurement mainly includes the pendulum instrument method, dynamic rotary friction coefficient method, side-way force coefficient method, and continuous trailer friction coefficient method, which calculates the friction coefficient by measuring the frictional resistance of rubber blocks or tires on the road surface [1,5–9]. The surface texture information is directly related to the friction coefficient of the pavement. Exploring the correlation between the texture state of the pavement and the friction coefficient can be suggested for the selection of raw materials, the design of the mixture, and the construction technology of the asphalt pavement [10–13].

To overcome the drawbacks as mentioned, researchers conducted many non-contact measurements which include digital image method, CT(Computed Tomography) scanning, and laser scanning approaches [14–16]. For example, Masad E. developed the Aggregate Image Measurement System (AIMS) for capturing profile images of coarse aggregate from all view perspectives [17]. The aggregate morphology evaluation index was simultaneously proposed. The higher the index of the shape, angle, and surface texture of coarse aggregate is, the greater the dynamic modulus of asphalt mixture is. Subsequently, Xiao Yue [18] applied digital image technology to the collection of texture information of asphalt pavement, and they could reconstruct the macrotexture form of pavement based on the functional relationship between image gray value and elevation value. Moreover, the correlation between the results of texture depth measured by sand patch method and image method is good. CT scanning technology can not only obtain the internal texture information of the pavement, but also display the pavement texture data. However, the CT scanning cost is expensive, and the test range is relatively small and limited to laboratory tests [19–21].

With the development of laser sensing technology, high-precision laser scanning instruments are gradually used in the road industry. In general, the maximum scanning area is 107.95 mm × 72.01 mm [22]. Qian used an EXAscanTM hand-held laser scanner to collect the macrotexture of a 9 cm × 9 cm area of asphalt pavement at an interval of 0.4 mm and then applied the pavement texture model to ANSYS for evaluating the fatigue performance of pavement texture by contact calculation with simplified tire model [23]. Ames Engineering has developed a laser scanner SLP, which uses the principle of laser triangulation reflection to obtain the function of the fluctuation amplitude of the measured pavement surface changing with the moving distance. Wang et al. [24] developed a laser texture device to test the profile of different micro surfacing pavements and indicated that

the use of steel slag can benefit the skid resistance of pavement. However, considering the current measurement and evaluation of asphalt pavement texture, the low scanning accuracy limited to the scale of macro pavement texture still exists. Furthermore, the three-dimensional shape evaluation method of asphalt pavement texture needs to be thoroughly explored and supplemented.

Based on the above background, this paper furtherly upgrades the original laser equipment developed in our previous research [25] and designs a 3D laser scanner which can accurately collect 3D point cloud data and reconstruct the 3D model of asphalt pavement. Meanwhile, this paper proposes a texture section method to evaluate the spatial distribution of the three-dimensional texture of asphalt pavement surfaces, which can directly measure and evaluate the roughness of the pavement texture. Research results will provide technical support for the evaluation and numerical simulation of the skid resistance performance of the wearing course.

## 2. Introduction of 3D Laser Scanner

### 2.1. Equipment Composition

(1) Measurement system. The measurement system consists of a line laser and a power supply which is supplied by a 24 V battery. The effective measurement range is ±23 mm, and the maximum resolution is 0.05 mm. It can fully meet the needs of surface profile measurement not only for the general asphalt mixture surface layer (the nominal maximum aggregate size is 13 mm or 16 mm) but also even for the large void asphalt pavement, such as OGFC. The laser beam width is 25–39 mm, and the sampling frequency of the data acquisition can reach 64 kHz. The scanning of 300 mm × 300 mm area can be completed by five round trips within 2 min.

The laser ranging sensor utilizes the laser triangulation ranging method as the measurement principle. According to the principle, the linear laser beam is used to scan the rough road. The elevation difference relationship of different measuring points can be converted to draw the profile texture of the pavement surface according to the reflection data of the laser at different measuring points. Below is the mathematical expression of the test principle.

$$PS = \frac{count - (2^{16}/2)}{2^{16}/2} \times 2 \times sf \times 10^{-dp} \qquad (1)$$

where $PS$ is relative elevation, $count$ is measured by the laser device, $sf$ is the amplification coefficient, and $dp$ is the position coefficient of the laser device.

(2) Control system. The control system of the laser profile detector is composed of the gantry frame (see Figure 1) that fixes the laser head, the DC servo motor and control box that controls its movement, and the transformer power supply box. The gantry frame adopts a high-precision linear module, which has high precision and can be controlled at about 0.001 mm. As a bearing platform for lasers and servo motors, it has the characteristics of high strength and good stability.

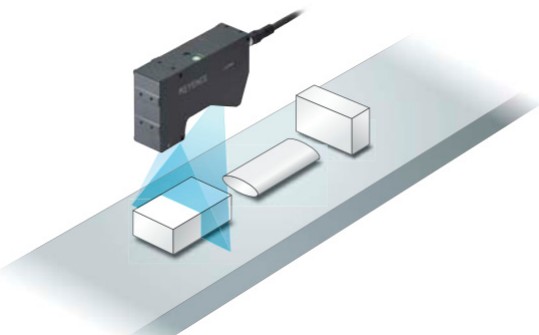

**Figure 1.** Schematic of 3D laser scanner.

(3) Data acquisition system. Regarding the data acquisition system of laser profile detector, the laser is connected to the computer, and the measurement data of the laser is saved to the computer in real time through the special software NAVIGATOR (Ver 2.00, KEYENCE Corporation; Tokyo, Japan) saved as a matrix data file.

(4) Display system. The laser scanner display system is LJ-Observer (Ver 1.00, KEYENCE Corporation; Tokyo, Japan) special software, with 2D and 3D display functions. The scanned profile line and the rendered 3D model can be displayed in real time to provide intuitive information for testers. However, limited by the width of the line laser beam, the display system can only display one scan of point cloud data at a time, and later it needs to be stitched into a complete specimen image with the help of MATLAB (Ver 9.1,2016, MathWorks company; MA, USA) software.

(5) Data analysis system. Since this equipment is independently developed, the data processing and index calculation are mainly programmed and calculated through the MATLAB (Ver 9.1,2016, MathWorks company; MA, USA) software platform. For the content of this part, see Sections 2.2, 3 and 4 of the manuscript.

### 2.2. Testing and Calibration of Equipment

(1) Interpolation processing of raw data

In the process of laser scanning, due to the unevenness of the pavement surface and the triangular reflection angle of the laser, there must exist a blind area in the laser scanning data due to the occlusion of the protruding texture of the road surface and the gap between the textures of the road surface in partial locations [26], which causes data to be missing in the area, usually represented by infinitely small negative values. The data missing not only affects the three-dimensional reconstruction effect of pavement morphology but also directly interferes with the data analysis results. Moreover, for missing data, interpolation is needed. The main processing methods of missing data interpolation are linear interpolation, Lagrange interpolation, nearest-neighbor interpolation, B-spline interpolation, and so on [27–29].

Linear interpolation mainly uses two values which are the closest to the missing data, and there is the calculation method [30]:

$$y_{ij}^* = y_{im} + \frac{y_{in} - y_{im}}{n - m}(j - m) \tag{2}$$

where, $y_{ij}^*$ is the estimated value of missing data $y_{ij}$; $y_{in}$, $y_{im}$ are the two nearest non-missing data; and $m \leq j \leq n$.

Specifically, Lagrange interpolation is based on the linear interpolation, but due to the multiple division and multiplication operations, the computation complexity is relatively high. Notably, the nearest-neighbor interpolation mainly adopts the similarity characteristics of adjacent nodes, which is apparently suitable for the small data set with missing data. Comparatively, the B spline interpolation approach can be described as a smooth curve passing through known key points, which is not suitable for the actual irregular aggregate texture and mutation characteristics of cracks.

In the actual tire and road contact, the automobile tire mainly contacts the top part of the road surface texture, where it is difficult to touch the depression area and texture cracks. It mainly provides road water storage and drainage channels and thereby reduces the thickness of the road surface water film. Therefore, based on the research background of this paper, the linear interpolation method is mainly used to deal with the missing points of laser data, and it is shown in Figure 2.

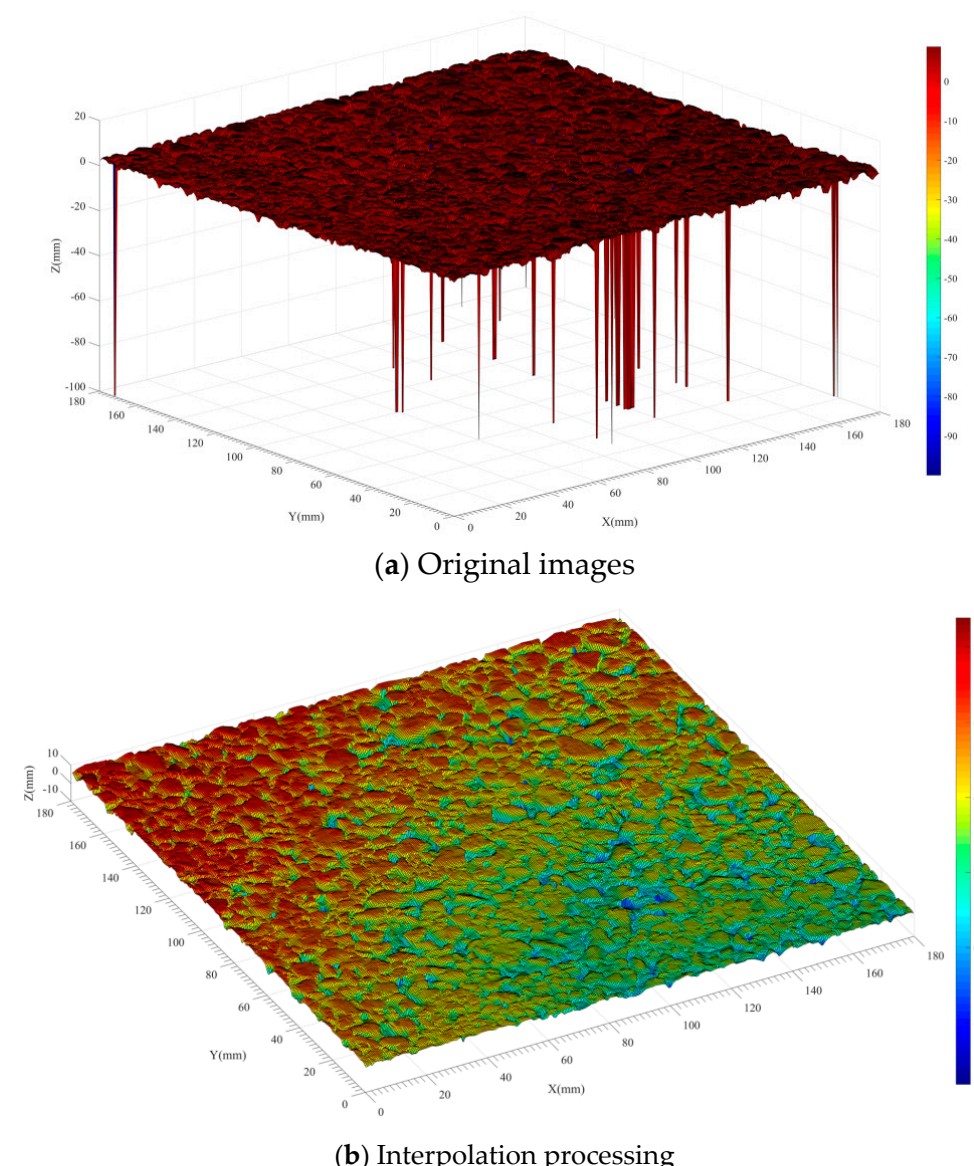

(**a**) Original images

(**b**) Interpolation processing

**Figure 2.** Data interpolation processing; (**a**) Original images; (**b**) Interpolation processing.

(2) Obliquity correction

Due to the unevenness of the road surface during the actual scanning process of the equipment, the laser scanning plane is not completely parallel to the road surface plane, so it is necessary to correct the obliquity of the scanning data. The existing research uses the least-squares method to correct the obliquity of a single profile of road surface [31]; however, it is obviously not applicable to irregular 3D road scanning data. Therefore, this study proposes the use of the least-squares plane fitting method for three-dimensional topography point cloud data slope correction. Obviously, if we take the row number x and column number y of the obtained matrix data as independent variables and the relative elevation value $z$ as dependent variable, then the fitting plane equation is $z^* = Ax + By + C$. The difference between the original shape matrix and the fitted plane, which is $z = z_0 - z^*$, is the modified three-dimensional matrix, and it is shown in Figure 3.

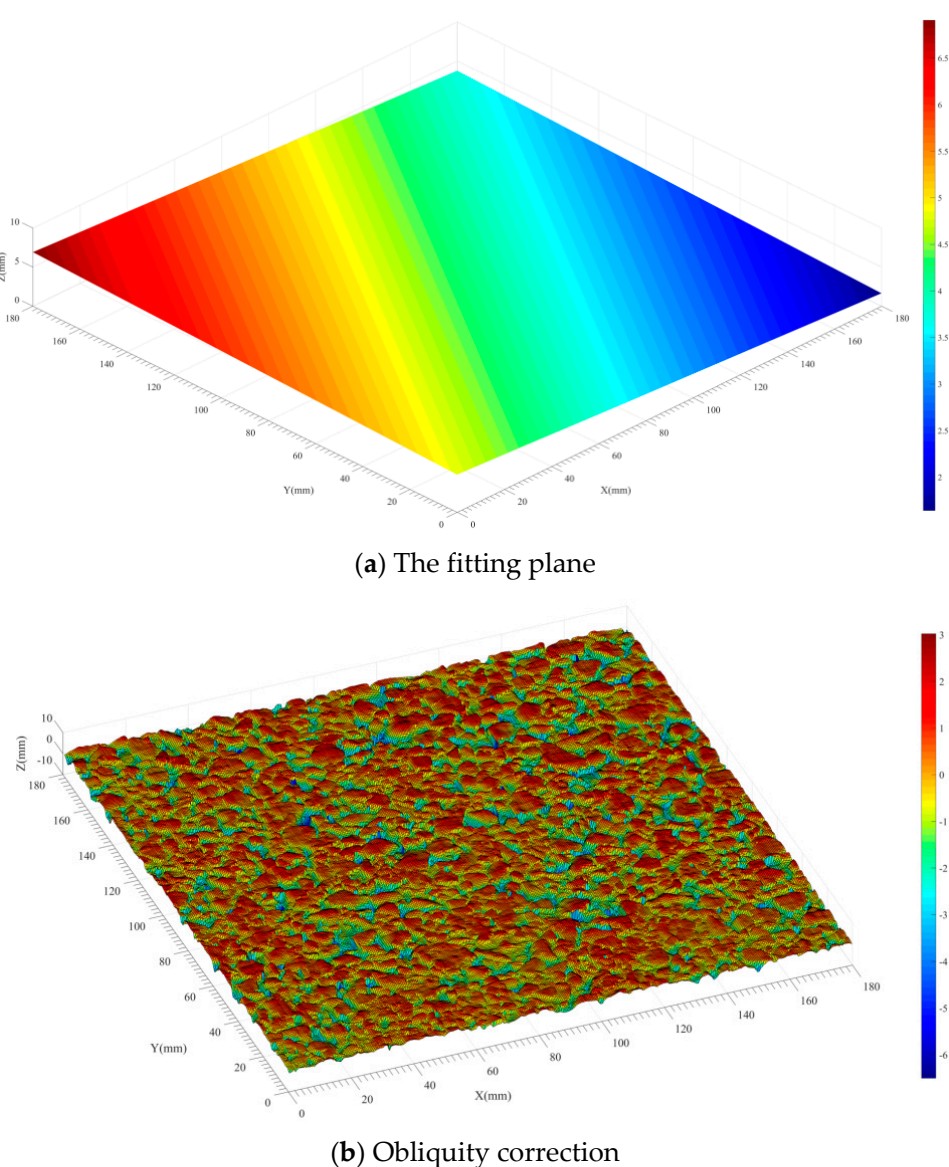

(**a**) The fitting plane

(**b**) Obliquity correction

**Figure 3.** Treatment of obliquity correction; (**a**) The fitting plane; (**b**) Obliquity correction.

(3) Calibration of equipment and error analysis

In order to ensure the reliability of the effective measurement range and accuracy of the scanner, it is necessary to calibrate and verify the equipment. The provisions of "geometrical product specification (GPS) Length Standard GB/T 6093-2001" in China for calibration blocks of different grades are used as well as the maximum accuracy of this laser scanning equipment. The self-designed calibration block is shown in Figure 4 and processed according to the three-level standard of the measurement block, so that the limit deviation of the total length of 120 mm is controlled at ±3 μm. The limit deviation of 5 mm single step height is controlled at ±1.2 μm. The error analysis of the test results was carried out, and the deviation diagram of the scanning results was plotted as shown in Figure 5. The 30 mm step width limit deviation is controlled at ±1.6 μm. The equipment can accurately obtain the profile shape of the object surface, and the proportion of the scanning deviation exceeding 95% is concentrated in the range of ±0.01 mm, which proves that the single measurement accuracy of the laser sensor and the mobile system can be controlled within 0.02 mm.

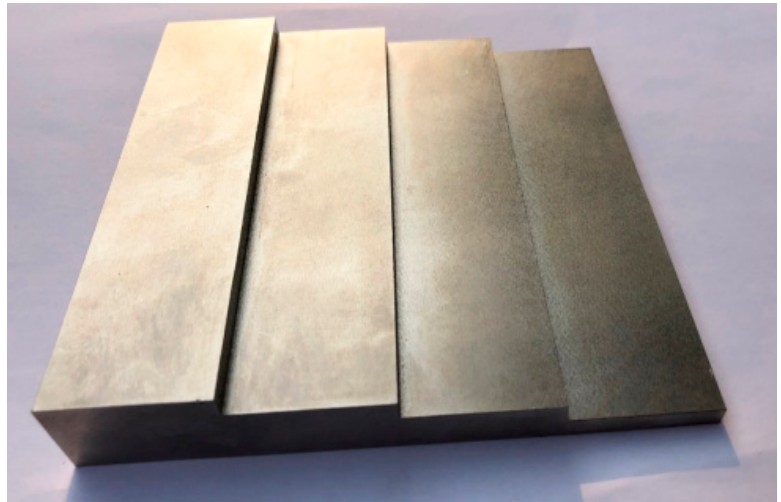

(**a**) Calibration test block

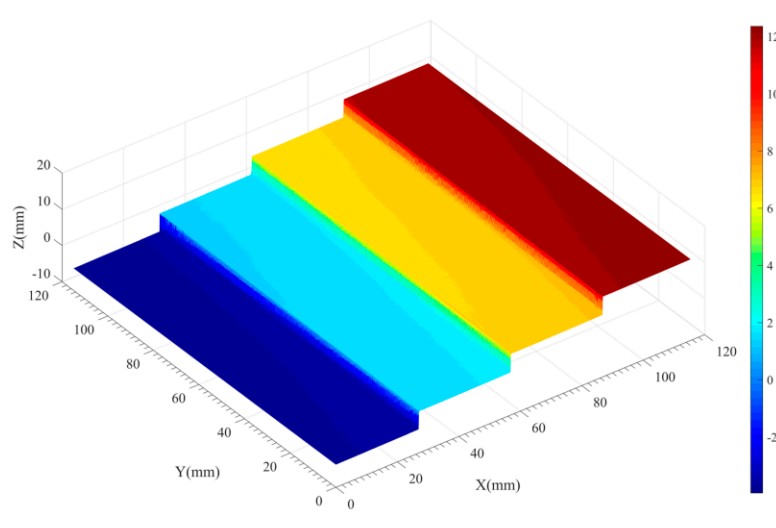

(**b**) Picture of 3D reconstruction

**Figure 4.** Design and scanning of calibration block; (**a**) Calibration test block; (**b**) Picture of 3D reconstruction.

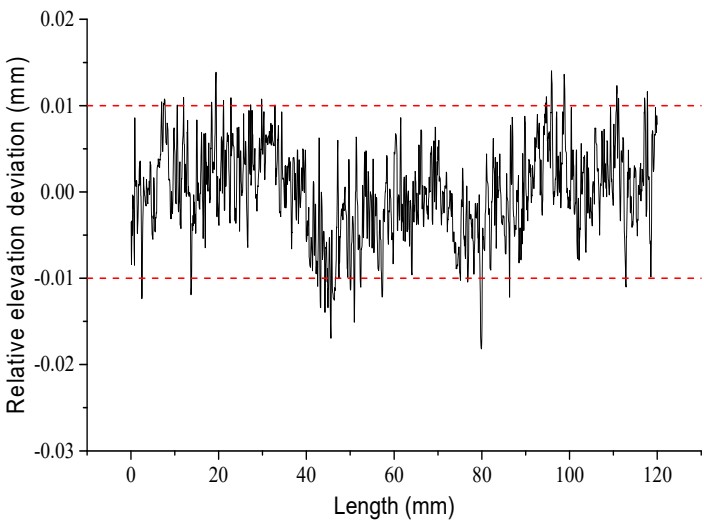

**Figure 5.** Measurement deviation.

## 3. Asphalt Mixture Design and Tests

### 3.1. Asphalt Mixture Design

The surface roughness of asphalt pavement texture is closely related to mixture type and gradation. Therefore, there are four wearing course textures in this study: skeleton-density asphalt concrete GAC-16, stone matrix asphalt SMA-13, asphalt mixture with highly asphalt content FAC-10, and ultra-thin wearing course GT-8. Among them, GAC, SMA, and FAC adopt SBS modified asphalt (PG76-22) produced by Shell, and the technical indicators are shown in Table 1. GT-8 adopts high viscosity polymer composite modified asphalt (PG 88-22), and the technical indicators are shown in Table 2.

**Table 1.** Technical indicators of SBS modified asphalt.

| Technical Indicator | Technical Requirement | Testing Result | Single Assessment |
|---|---|---|---|
| Penetration 25 °C, 100 g, 5 s, 0.1 mm | 40–60 | 52 | qualification |
| Penetration index PI    Min | +0.0 | +0.14 | qualification |
| Ductility 5 °C, 5 cm/min, cm    Min | 20 | 35 | qualification |
| Softening point $T_{R\&B}$ (°C)    Min | 75 | 80.0 | qualification |
| Flash point (°C)    Min | 230 | 333 | qualification |
| Solubility (%)    Min | 99 | 99.8 | qualification |
| Storage stability *:163 °C, 48 h, softening point difference °C    Max | 2.5 | 1.0 | qualification |
| Elastic recovery 25 °C, %    Min | 90 | 98 | qualification |
| Kinematic viscosity (Pa·s)    135 °C   Max | 3 | 2.78 | qualification |
| 165 °C | / | 0.78 | |

* This index refers to the maximum softening point deviation of asphalt materials stored in an oven at 163°C for 48 h.

**Table 2.** Technical indicators of high-viscosity polymer composite modified asphalt.

| Technical Indicator | Technical Requirement | Testing Result | Tingle Assessment |
|---|---|---|---|
| Penetration   25 °C, 100 g, 5 s, 0.1 mm | 30–50 | 38 | qualification |
| Penetration index PI    Min | +0.0 | +0.17 | qualification |
| Ductility 5 °C, 5 cm/min    Min | 20 | 22 | qualification |
| Softening point $T_{R\&B}$    Min | 75 | 98.0 | qualification |
| Flash point    Min | 230 | 320 | qualification |
| Solubility (%)    Min | 99 | 99.8 | qualification |
| Storage stability *: 163 °C, 48 h, softening point difference °C    Max | 2.5 | 2.1 | qualification |
| Elastic recovery 25 °C, %    Min | 95 | 99.5 | qualification |
| 60 °C complex shear modulus G *(kPa)    Min | 12 | 14.5 | qualification |
| Dynamic viscosity 60 °C (Pa·s)    Min | 500,000 | 828,751 | qualification |

* This index refers to the maximum softening point deviation of asphalt materials stored in an oven at 163°C for 48 h.

The coarse aggregate is diabase, and the technical indicators are shown in Table 3. The fine aggregate is limestone, and the technical indicators are shown in Table 4. The filler is fine-grained limestone mineral fines.

16 rut plate (300 mm × 300 mm × 50 mm) specimens were formed. The ratio of asphalt to stone of GAC-16, SMA-13, FAC-13, and GT-8 are 4.7%, 6.1%, 5.3%, and 6.5%, respectively. The gradation of the four asphalt mixtures are shown in Table 5.

**Table 3.** Test results of coarse aggregate performance evaluation.

| Test Index | Technical Requirement | | Testing Result | Single Assessment |
|---|---|---|---|---|
| | Unit | Design Requirement | | |
| Crushing value of stone | % | ≤20 | 10.2 | qualification |
| Los Angeles abrasion loss | % | ≤28 | 12.3 | qualificatio n |
| Apparent relative density | — | ≥2.60 | 2.890 | qualification |
| Water absorption rate | % | ≤2.0 | 0.63 | qualification |
| Ruggedness | % | ≤12 | 3.4 | qualification |
| Flat elongated particles content (mixture) | % | ≤15 | — | — |
| The particle size is more than 9.5 mm | % | ≤12 | 7.7 | qualification |
| The particle size is less than 9.5 mm | % | ≤18 | — | — |
| Water washing method particles content <0.075 mm | % | ≤1.0 | 0.2 | qualification |
| Soft stone content | % | ≤3 | 1.4 | qualification |
| Adhesion with modified asphalt | level | 5 | 5 | qualification |
| Polishing value | PSV | ≥42 | 44 | qualification |

**Table 4.** Test results of fine aggregate evaluation.

| Test Index | Technical Requirement | | Sample Specification | 0–3 mm |
|---|---|---|---|---|
| | Unit | Highway First Class Highway | Testing Result | Single Assessment |
| Apparent specific gravity | — | ≥2.50 | 2.732 | qualification |
| Ruggedness (the part >0.3 mm) | % | ≤12 | 2.4 | qualification |
| Sand equivalent | % | ≥65 | 72 | qualification |
| Methylene blue number | g/kg | ≤2.5 | 1.5 | qualification |
| Angularity (flow time) | s | ≥30 | 39 | qualification |

**Table 5.** Gradation of four asphalt mixtures.

| Grading Type | Percentage Pass Rate of Sieve Hole (mm)/% | | | | | | | | | | | |
|---|---|---|---|---|---|---|---|---|---|---|---|---|
| | 26.5 | 19 | 16 | 13.2 | 9.5 | 4.75 | 2.36 | 1.18 | 0.6 | 0.3 | 0.15 | 0.075 |
| GAC-16 | 100 | 100 | 98.7 | 86.2 | 59.2 | 33 | 23.7 | 18.2 | 14.1 | 11.2 | 7.8 | 5.4 |
| SMA-13 | 100 | 100 | 100 | 95.9 | 59 | 27.3 | 22.5 | 18.9 | 16.4 | 14 | 12.8 | 10 |
| FAC-10 | 100 | 100 | 100 | 100 | 98.4 | 37.5 | 33.1 | 26.3 | 19.8 | 17.3 | 15.7 | 11.2 |
| GT-8 | 100 | 100 | 100 | 100 | 100 | 48.7 | 19.8 | 14.9 | 11.4 | 8.6 | 7.8 | 5.8 |

*3.2. Tests*

(1) Prepare the rutting plate specimens of different asphalt mixtures, and test the average texture depth of the specimens by sand patch method.

(2) Place specimen under 3D laser scanner, adjust laser sensor to optimum test height (80 mm from specimen surface), start equipment so that the laser sensor head scans back and forth 10 times to finish the scanning of every specimen, and output relative elevation point cloud data.

(3) Complete the splicing, linear interpolation and obliquity correction of scanned data on MATLAB, and generate pavement texture surface model shown in Figure 6.

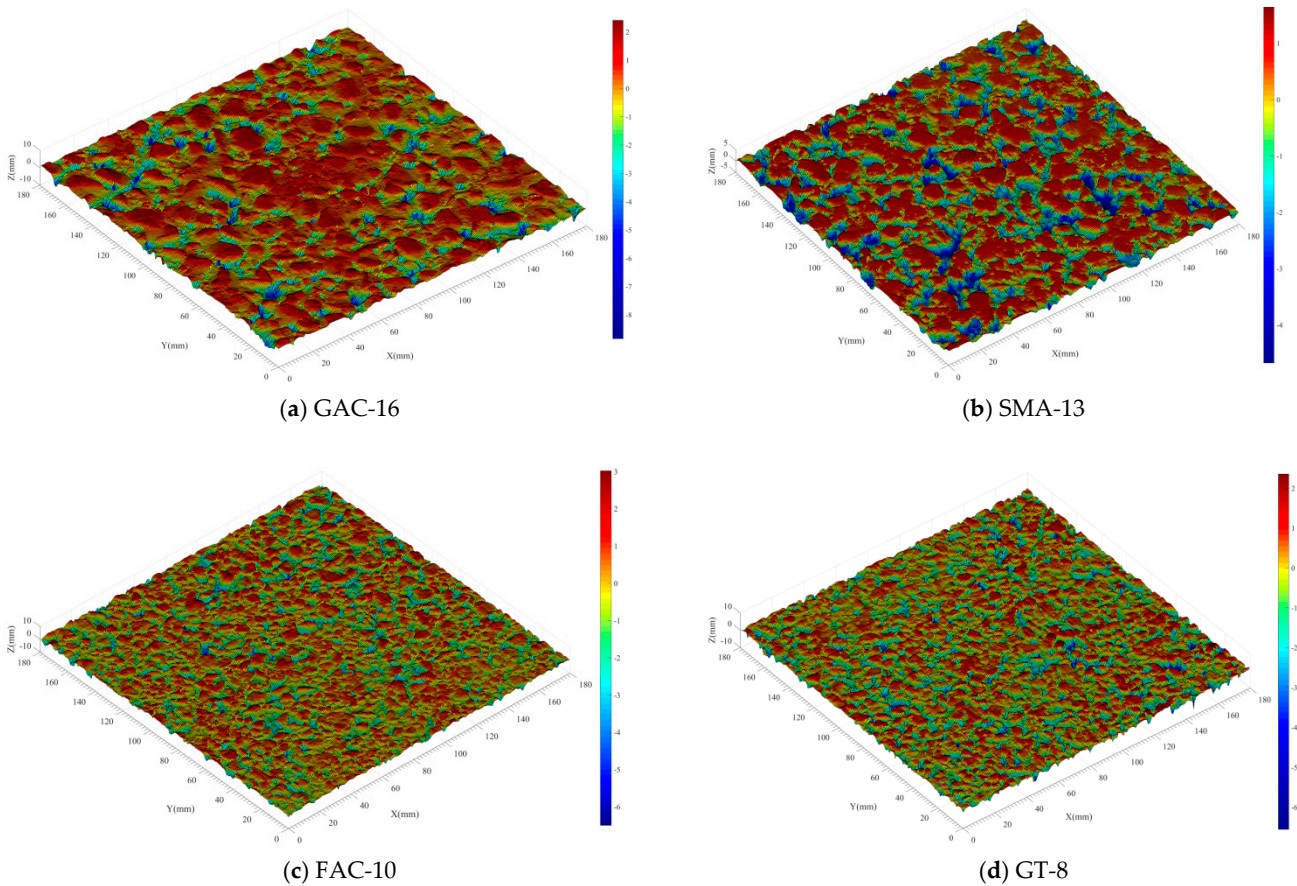

**Figure 6.** Laser scanning results of different pavements; (**a**) GAC-16; (**b**) SMA-13; (**c**) FAC-10; (**d**) GT-8.

## 4. Assessment of Texture Features of Asphalt Pavement

The accuracy of the French ISO 13473-1 vehicle laser sensor for the measuring depth of texture ranges from 0.5 mm to 1.0 mm. ASTM E2147 recommended the use of arc texture tester with 0.87 mm interval to collect a ring texture profile whose radius is 142 mm; then, the average section depth of pavement is calculated. At the same time, microtexture wavelength is less than 0.5 mm, and its amplitude ranges from 0.001 mm to 0.5 mm according to the scale definition of road surface texture by the International Road Association. The macrotexture wavelength ranges from 0.5 mm to 50 mm, and its amplitude ranges from 0.1 mm to 20 mm. Therefore, this study chooses 0.5 mm as sampling interval to obtain macrotexture and chooses 0.05 mm interval to obtain microtexture distribution.

According to previous studies, the contact area of tires is generally 160–180 mm [25]. At the same time, considering the computing performance of ordinary computers, this study focuses on the selection of a 180 mm × 180 mm area in the middle of the rut plate specimen for calculation and analysis. The maximum value of the measured area approaches the value of 32,400 mm$^2$.

### 4.1. Evaluation Index of Road Surface Profile

For the characterization of pavement texture morphology, the profile of the texture is mainly described, and the root mean square deviation of profile is a commonly used roughness characterization index [32]. Its calculation formula is as follows:

$$R_q = \sqrt{\frac{1}{N}\sum_{i=1}^{N} y^2(x_i)} \qquad (3)$$

where $R_q$ is the root mean square deviation of profile; $N$ is the number of measurement points of the profile; and $y(x_i)$ is the relative elevation of each measurement point.

The relative elevation point cloud distribution of 3D surface is obtained based on the lowest point of pavement scanning data. The profile of laser scanning point cloud of four asphalt pavements is extracted. About 260 profiles on the surface of asphalt mixture specimens were extracted with 1 mm spacing (40 mm length at both ends was deleted considering the segregation of the specimen edges), and the root mean square deviation of profile of four types of pavement is calculated respectively (Figure 7). Table 6 is a summary of the root mean square deviation results of Figure 7.

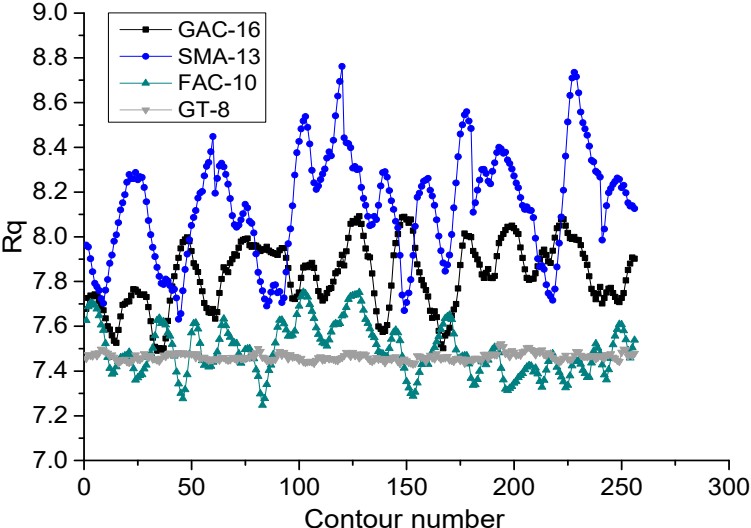

**Figure 7.** Root mean square deviation distribution of profile of different pavements.

**Table 6.** Summary of root mean square deviation results of profile of different pavements.

| Pavement Types | Minimum Value | Maximum Value | Mean Value | Standard Deviation |
| --- | --- | --- | --- | --- |
| GAC-16 | 7.490 | 8.092 | 7.830 | 0.148 |
| SMA-13 | 7.632 | 8.761 | 8.137 | 0.247 |
| FAC-10 | 7.247 | 7.753 | 7.494 | 0.112 |
| GT-8 | 7.432 | 7.522 | 7.463 | 0.016 |

As indicated from Figure 7 and Table 6, the root mean square deviation level of surface profile of different asphalt pavements is different. Among them, the root mean square deviation of SMA-13 is the largest, followed by GAC-16 and FAC-10, whereas GT-8 is the smallest, which is mainly related to the maximum nominal particle size and mineral aggregate gradation of asphalt mixture. Even for the same road type, the root mean square deviation of profile calculated by different surface profiles fluctuates greatly. The fluctuation of the root mean square deviation of the SMA-13 ranges from 7.632 to 8.761, and the standard deviation is 0.247, whose fluctuation is the greatest. The fluctuation of the root mean square deviation of GAC-16 ranges from 7.49 to 8.092, the standard deviation is 0.148, and the fluctuation level is slightly smaller than that of SMA-13. The reason is that, in the molding process of asphalt mixture, the random distribution of aggregate particles, coupled with the rise of asphalt mortar, caused large heterogeneity of the spatial distribution of asphalt pavement texture. Moreover, the proportion of coarse aggregates (>4.75 mm) adopted in the SMA-13 mixture was significantly higher than that of GAC-16, which made the surface texture of SMA-13 rougher and caused the distribution of coarse aggregate particles to be more complex. For the GT-8 with the smallest nominal particle

size, the surface profile index fluctuates the least, which is mainly related to the smaller nominal particle size. The coarse aggregate particles are mainly 2.36–4.75 mm and 4.75–8 mm. The size span of coarse aggregates in GT-8 is small, and the proportion is high. Therefore, it can form a more uniform surface texture.

Otherwise, although the two-dimensional profile indexes of different asphalt pavements can evaluate the profile characteristics of road surface texture to a certain extent at the statistical level, they also have the problem of insufficient representation of a single profile curve. However, for the roughness of the road, they pay more attention to the wheel track area of the road, which is the three-dimensional interface between the tire and the road [25], so it is necessary to evaluate the roughness of asphalt pavement from a three-dimensional perspective.

### 4.2. Simplified Representation Model of Pavement Texture

The pavement texture depth can reflect the drainage capacity of the macrotexture of asphalt pavement, but it cannot truly reflect the three-dimensional of the pavement texture. The three-dimensional surface model of pavement texture reconstructed by laser scanning equipment is used, and the plane at the highest point of the model is used as the top surface of the texture to extract the macrotexture morphology of the pavement at different depths. Taking the FAC-10 pavement model as an example, the three-dimensional distribution of pavement macrotexture at different depths is shown in Figure 8. In order to make it more intuitive, a 90 mm × 90 mm range is selected to display. It can be clearly seen that the macrotexture of the road surface changes from shallow to deep, and the distribution of the texture changes from sparse to dense.

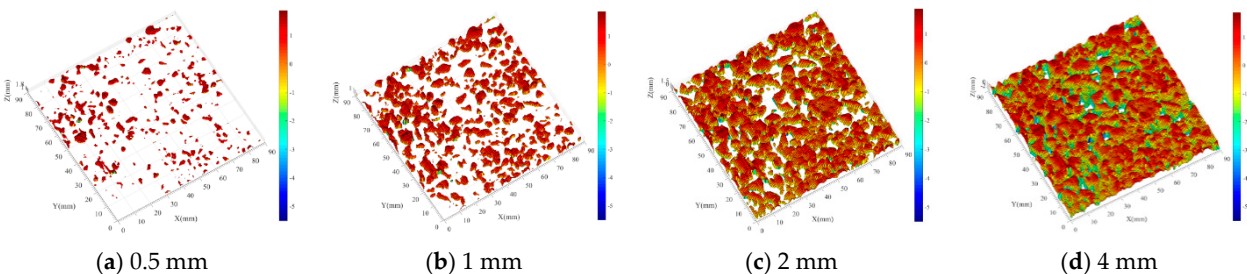

| (**a**) 0.5 mm | (**b**) 1 mm | (**c**) 2 mm | (**d**) 4 mm |

**Figure 8.** Pavement texture distribution at different depths; (**a**) 0.5 mm;(**b**) 1 mm; (**c**) 2 mm; (**d**) 4 mm.

According to Hertz's contact theory, single-point rough bodies are usually simplified as spherical, cylindrical, and conical models [33,34] shown in Figure 9. According to the geometric characteristics of the simplified model, the mathematical relationship between the profile area and the cutting depth of the model can be established.

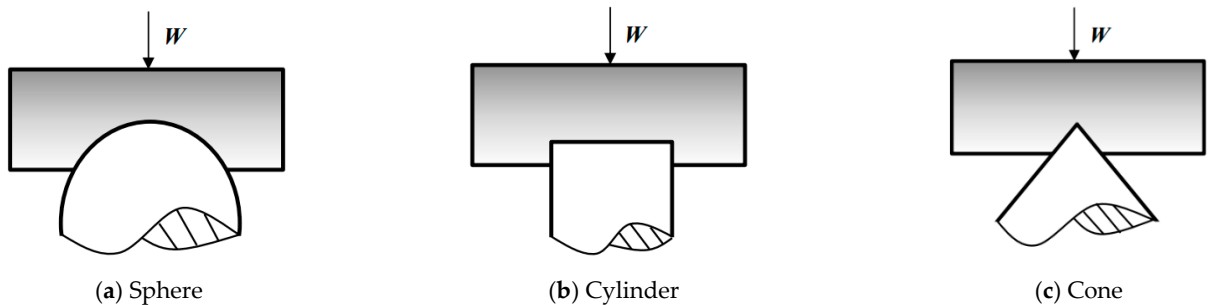

| (**a**) Sphere | (**b**) Cylinder | (**c**) Cone |

**Figure 9.** Rough peak model; (**a**) Sphere; (**b**) Cylinder; (**c**) Cone.

For the sphere model, there is the texture section area at different depths.

$$A = \pi \times r^2 = \pi \times \left(2dh - h^2\right) \tag{4}$$

where $\pi$ is the circumference, $r$ is the section circle radius of a certain depth, $d$ is the radius of sphere model, and $h$ is the section depth.

For the cylindrical model, there is the texture section area at different depths:

$$A = \pi \times r_0{}^2 \tag{5}$$

where $r_0$ is the radius of the cylinder top surface.

For the cone model, there is the texture section area at different depths:

$$A = \pi \times r^2 = \pi \times \tan^2\left(\frac{\theta}{2}\right) \times h^2 \tag{6}$$

where $\theta$ is the top angle of the cone.

The schematic diagram of the area change curve of the macrotexture section at different depths of different simplified rough peak models was drawn and is shown in Figure 10.

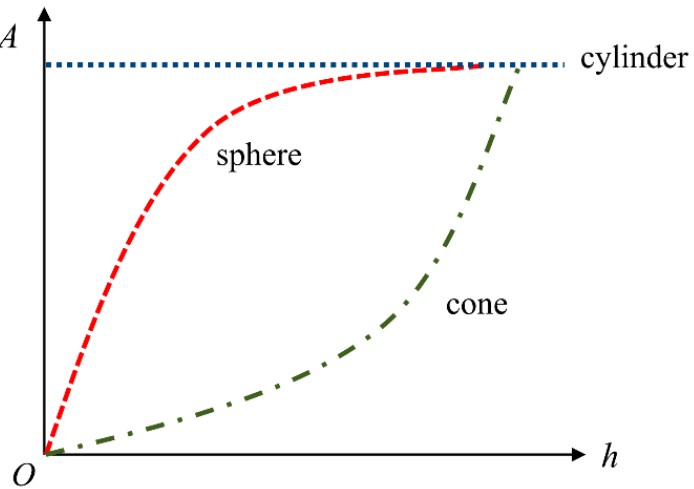

**Figure 10.** Three-dimensional variation curves of different simplified models.

According to the principle of skid resistance [2], the macrotexture can provide the blocking friction between the tire deformation and the drainage channel. The sharpness of its texture can influence the stress concentration of tire contact and the deformation degree of tread rubber. The section area of three dimensional surface model can be used to characterize the pavement macroscopic features under different depths. Then, the construction model at a certain depth can be intercepted layer by layer, and the projection area of the layer texture can be counted. Finally, the texture section area change curve at different depths can be drawn. Taking FAC-10 as an example, the calculation results are shown in Figure 11.

When the model is intercepted at a depth of 1–2 mm, the area variation curve of the macrotexture section is close to that of the conical model. In the middle curve section, the change of texture section area changes from fast to slow, and its change trend is like that of the spherical model curve. When the intercepted depth is large, the area of texture section does not change significantly, which can be regarded as the cylinder model curve. Therefore, for the pavement texture composed of different coarse aggregates, its three-dimensional shape cannot be characterized by a single simplified model, while using different simplified models to characterize the texture feature at different depths is an effective method.

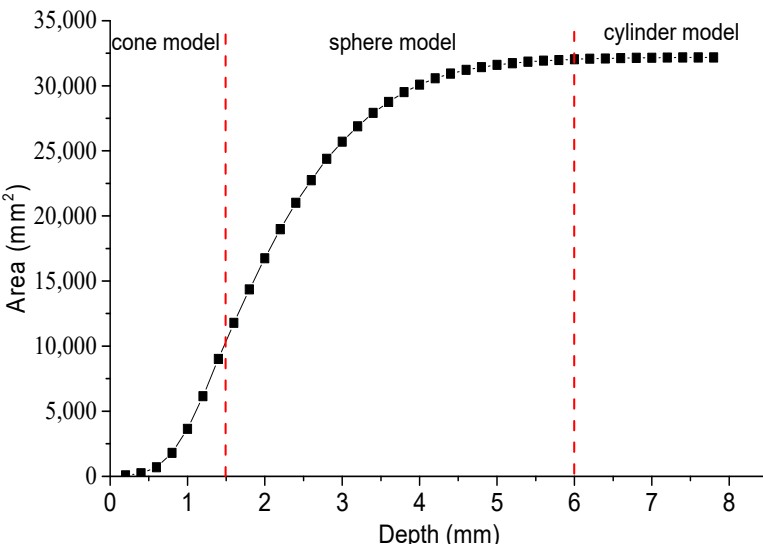

**Figure 11.** Area variation curve of macrotexture section at different depths.

The macrotexture morphology data of different types of asphalt mixture pavement are collected, and the macrotexture section area of different pavement depths is calculated, with its changing curve drawn and shown in Figure 12. It is easy to find that at the same depth, the size of the macro-texture section area is GT-8, FAC-10, SMA-13, and GAC-16 in turn, which shows that the skeleton with the smaller nominal particle size has a denser surface texture. The change trend of three-dimensional texture morphology of asphalt pavement with different gradation types is similar, and the changing curve of section area basically experiences the combination model of "cone + sphere + cylinder".

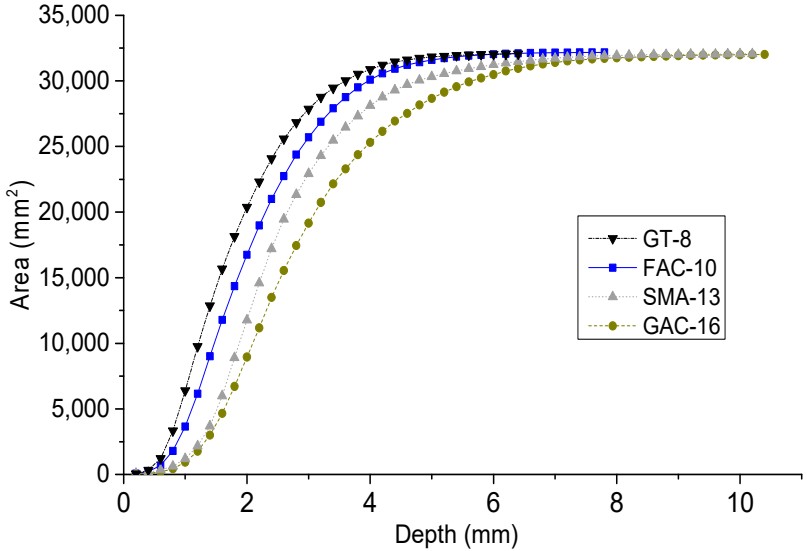

**Figure 12.** Texture section area change of different pavements.

### 4.3. Roughness Evaluation of Asphalt Pavement

The skid resistance of pavement is mainly provided by the macro and microtexture of pavement surface. Macrotexture includes protrusion height and angularity, and it is mainly related to mixture gradation design and coarse aggregate processing grain shape. The microtexture is mainly distributed on the aggregate crushing surface and mainly determined by the aggregate lithology and processing technology. Asphalt pavement is usually firstly polished under the action of tires, which is mainly reflected in the gradual wear of asphalt film and the prominent microtexture of aggregate surface. Then, it is cut

under the normal load and tangential shear force of the tire, and the friction coefficient of the road decreases rapidly at this stage. Finally, the angularity and height of the macrotexture of the pavement gradually wear under the impact and wear of the vehicle dynamic load in a long period of service life. In addition, the secondary kneading and compaction occurring in the wheel track tape area of the pavement mainly causes the decline of texture depth [35].

Because the aggregate microtexture is defined as the wavelength and height of 0.001 mm–0.5 mm, the laser scanning measurement system is adopted. The plane scanning resolution of the equipment is set to 0.05 mm, and the measurement accuracy is set to 0.01 mm. The microtextures of different types of asphalt pavements are respectively obtained. Taking the FAC-10 specimen as an example, the microtexture of the texture surface is shown in Figure 13. However, the microtexture features cannot be quantitatively analyzed and evaluated based on the microtexture images of the mixture surface. It is necessary to further process the micro-morphology data. According to related research, factors such as the height, sharpness, and density of the micro convex body contribute to the adhesion friction generated by tire-pavement contact [36]. However, due to the difficulty in defining and counting the particle adhesion of the micro convex body, there is still no unified conclusion on the microtexture evaluation of texture surfaces.

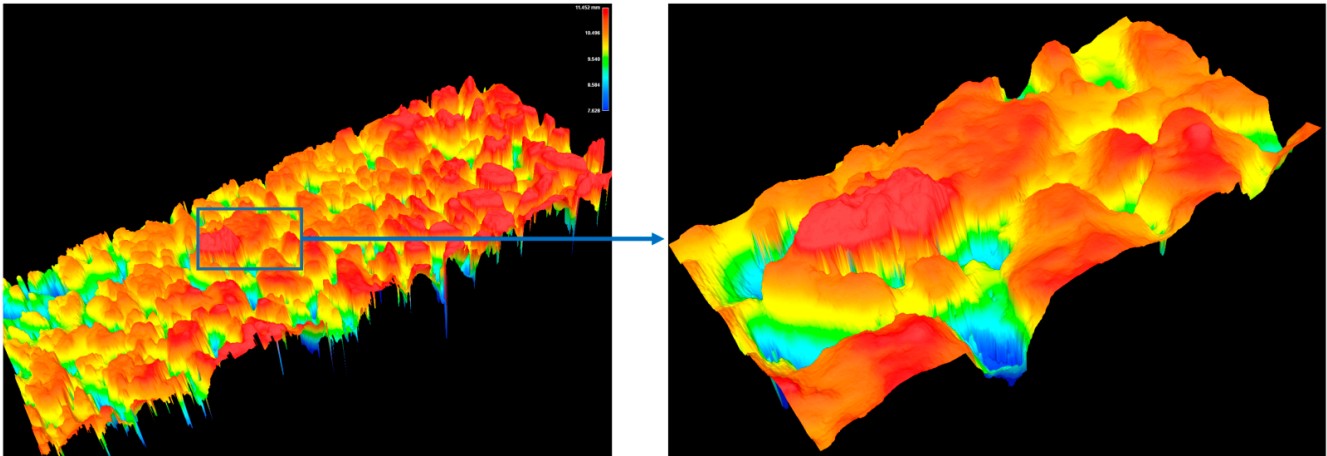

**Figure 13.** Microtexture scanning schematic.

Based on the three-dimensional surface model of microtexture, the three-dimensional surface area of microtexture and macrotexture at different depths is determined layer by layer, and then the distribution density of macrotexture and microtexture is evaluated. At present, the solution of three-dimensional surface area mainly includes integral method and projection method [37]. The integral method is generally applicable to more regular surface graphics, but the solution of surface equations is extremely difficult for irregular graphics. Conversely, the projection method is more suitable for irregular surface graphics [38].

The basic principle is that projected discrete spatial data points to *xoy*, *yoz*, and *zox* planes, respectively, and then the data points of the three planes can be represented as follows:

- *xoy* plane: $\{(x_1, y_1), (x_2, y_2), \cdots, (x_n, y_n)\}$
- *yoz* plane: $\{(y_1, z_1), (y_2, z_2), \cdots, (y_n, z_n)\}$
- *zox* plane: $\{(z_1, x_1), (z_2, x_2), \cdots, (z_n, x_n)\}$

Therefore, the polygon area enclosed by spatial data points in three planes is calculated as Equations (7)–(9).

$$A_{xoy} = \frac{1}{2}\sum_{i=1}^{n}(x_{i+1} + x_i)(y_{i+1} - y_i) \tag{7}$$

$$A_{yoz} = \frac{1}{2}\sum_{i=1}^{n}(y_{i+1} + y_i)(z_{i+1} - z_i) \tag{8}$$

$$A_{zox} = \frac{1}{2}\sum_{i=1}^{n}(z_{i+1} + z_i)(x_{i+1} - x_i) \tag{9}$$

In addition, the surface area is calculated as Equation (10).

$$A = \sqrt{A_{xoy}^2 + A_{yoz}^2 + A_{zox}^2} \tag{10}$$

Through writing a program in MATLAB (Ver 9.1,2016, MathWorks company; MA, USA) software to iterate the matrix data line by line to calculate the area indicators at different depths and taking four kinds of asphalt mixture as examples, the calculation results are shown in Figure 14. The surface texture distribution of asphalt pavement has an obvious scale effect. The microtexture scale is the smallest and the surface profile area is the largest, which is about 1.8–2.2 times of the macrotexture sectional area. It is mainly related to the number, height, and sharpness of the micro convex body. Due to its angularity and grain shape, the surface area of macrotexture is about 1.4 times that of the sectional area.

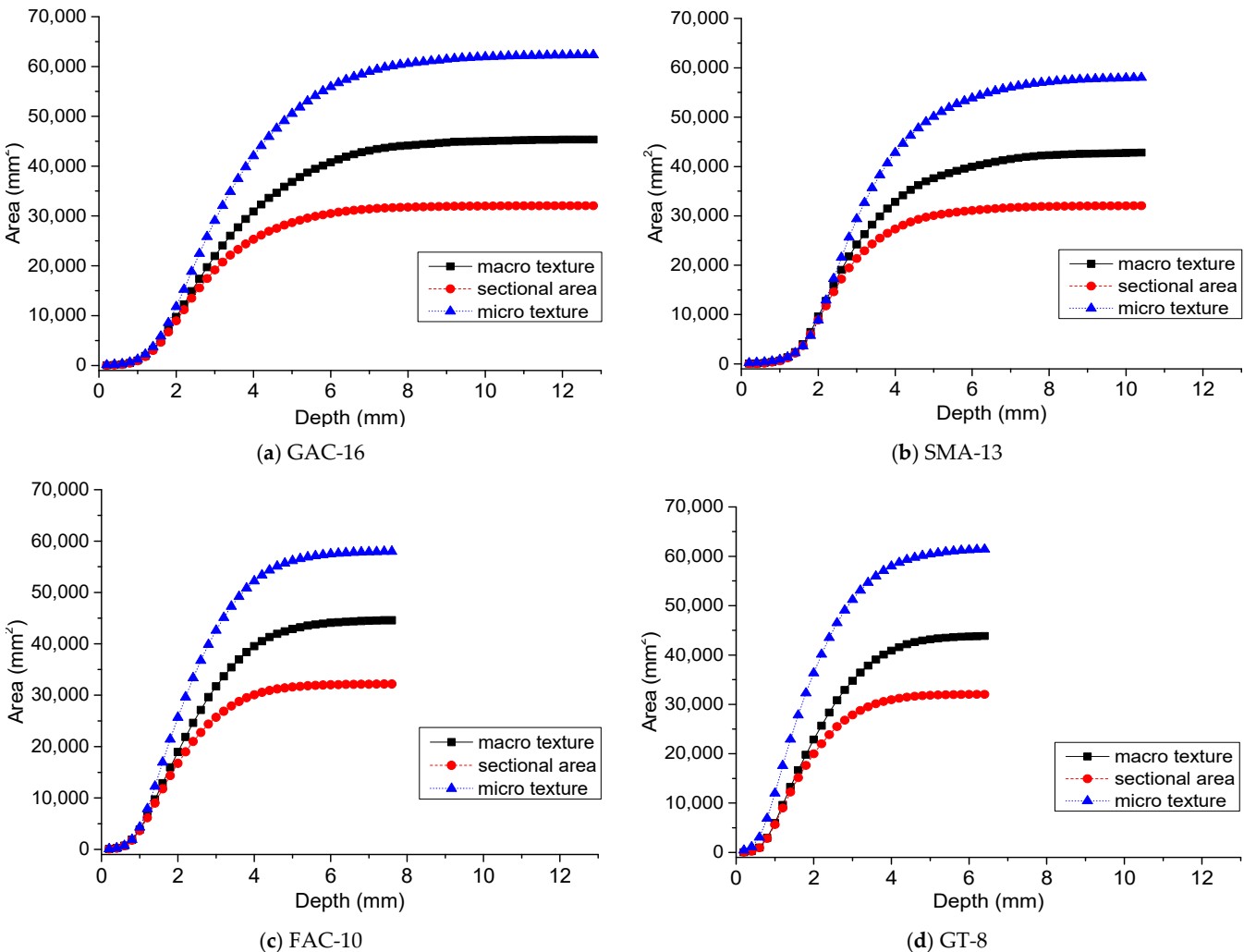

**Figure 14.** Different pavement texture distribution under different depths; (**a**) GAC-16; (**b**) SMA-13; (**c**) FAC-10; (**d**) GT-8.

In order to evaluate the density of pavement surface texture distribution better, it is necessary to standardize the calculation results. Based on the projection plane constructed

by three-dimensional surface, the macroscopic and microscopic surface areas in this range are calculated. The richer the road surface texture of the area is, the greater the surface area of the macrotexture is. The rougher the aggregate microtexture is, the larger the micro-texture surface area is. It uses the texture area in the unit datum plane to characterize the distribution density of pavement surface texture, which is defined as textured distribution density.

$$S = \frac{A_T}{A_P} = \frac{A_T}{a \times b} \tag{11}$$

where $S$ is the texture distribution density, including $S_{MA}$ and $S_{MI}$ indexes respectively. $S_{MA}$ is the macrotexture distribution density and $S_{MI}$ is the microtexture distribution density; $A_T$ is the macro or micro three-dimensional surface area, mm$^2$; and $A_P$ is the plane area of the texture section area (the horizontal projection area), which can be calculated by the edge length a and b of the measurement area, mm$^2$.

### 4.4. Correlation between Pavement Texture Density and Skid Resistance Performance

In order to verify the accuracy of the three-dimensional evaluation index of pavement texture based on the laser scanner measurement system, the texture depth of different types of asphalt mixture specimens was tested by the sand patch method, and the friction coefficient BPN was tested by the British Pendulum Tester. Results are shown in Figure 15.

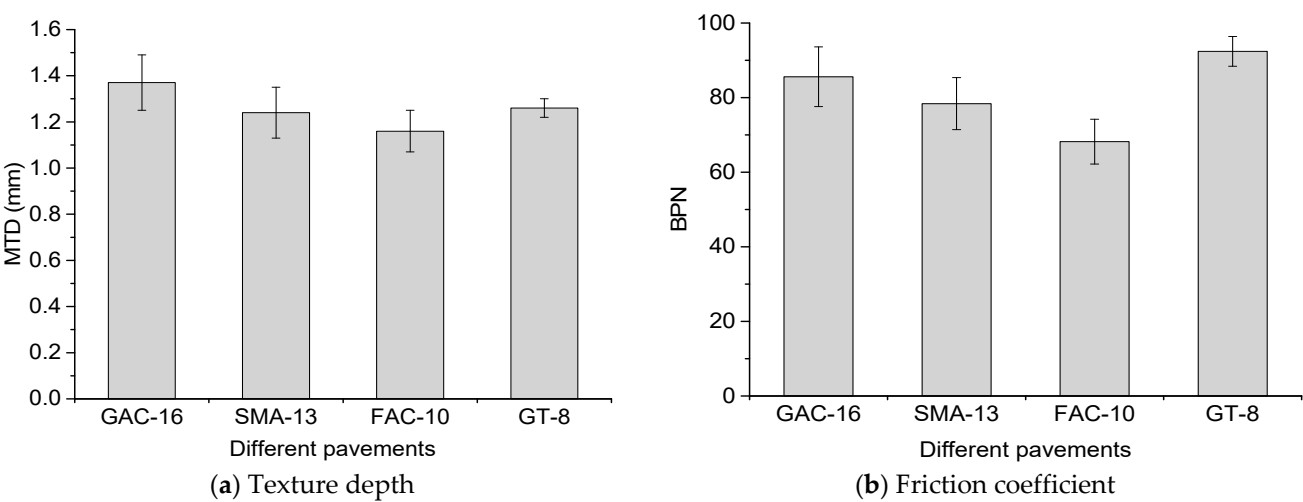

**Figure 15.** Texture depth and pendulum test results of different pavements; (**a**) Texture depth; (**b**) Friction coefficient.

The three-dimensional surface model was tested by the laser scanning method, and the $S_{MA}$ and $S_{MI}$ were calculated. From Figure 16, there are obvious differences between texture density of asphalt pavements. The $S_{MA}$ of different mixtures is about 1.4 while the $S_{MI}$ is above 1.8, which indicates that the microtexture of a coarse aggregate surface makes the surface texture rougher. The $S_{MI}$ of FAC-10 and SMA-13 is smaller, which is on account of that the thicker asphalt mortar cover the coarse aggregate surface part of the micro convex body texture. The GT-8 mixture with smaller nominal particle size has the highest microtexture density, and the coarse aggregate used in the test is the same lithology and the same quarry processed gravel. The microtexture on the aggregate surface can be regarded as uniform distribution. Therefore, it can be considered that the smaller the coarse aggregate texture is, the more compact its arrangement is, which is conducive to improving the density of microtexture distribution on pavement surface.

The correlation between $S_{MA}$ and MTD (mean of the four pavements) was established and shown in Table 7. The distribution density of macrotexture increases with the increase in the pavement texture depth, and their correlation coefficient is 0.838. The $S_{MA}$ value of different asphalt pavements is between 1.387–1.413, which is mainly related to the resolution of the set equipment. From the perspective of the correlation model, the index

has a good correlation with the texture depth index, which can reflect the macrotexture level of the road surface to a certain extent.

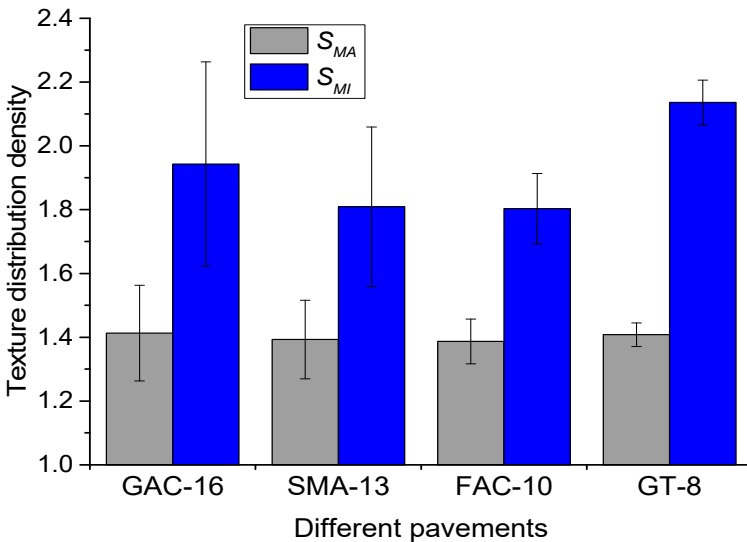

**Figure 16.** Texture distribution of different mixtures.

**Table 7.** Correlation between pavement texture density and skid resistance performance.

| Type | *a* | *b* | $R^2$ |
| --- | --- | --- | --- |
| $S_{MA}$ and MTD | 0.0646 | 1.3143 | 0.838 |
| $S_{MI}$ and BPN | 0.0102 | 1.0491 | 0.846 |

Due to the small scale of microtexture, the traditional surface texture test method has difficultly directly measuring the microtexture of asphalt pavement. At present, the British Pendulum Tester is used to indirectly reflect the microtexture level of pavement. Therefore, this study mainly adopts the British Pendulum Tester to obtain the surface friction coefficient of specimens. Moreover, in order to minimize human error, all specimens in this paper were tested by professionals in strict accordance with the test procedures. The relationship between $S_{MI}$ and BPN (mean of the four pavements) is established and shown in Table 7. The correlation coefficient between the pendulum friction coefficient and the surface microtexture distribution density of the specimen is 0.845, which has a good correlation. The $S_{MI}$ can well characterize the richness level of microtexture on the asphalt pavements. Matúš Kováč et al. used a high-precision laser scanner to detect the texture of the 30 mm × 120 mm area on the road surface and found that a single macrotexture or microtexture index has relatively poor correlation with the friction coefficient [39]. This finding is consistent with the previous attempts of this study, and the reason is that the test results of different specimens have relatively large fluctuations, which are mainly related to the difference in the test range of the two methods and the effective contact surface of the rubber block. However, the number of samples is small, and the test is mainly carried out on the specimens covered by the asphalt film. The three-dimensional morphology index that considers the macro and microtexture proposed by this research has a certain characterization significance.

## 5. Summary

(1) The fluctuation of root mean square deviation of the surface profiles of different asphalt pavements is quite different. The fluctuation of SMA-13 and GAC-16 profiles is the largest, followed by FAC-10, while GT-8 had the least fluctuation, which is mainly affected by the nominal particle size and coarse aggregate ratio of the asphalt mixture. The uneven

distribution of the pavement macrotexture makes it difficult for the road surface profile to fully characterize the roughness of the asphalt pavement.

(2) Using a self-developed 3D laser scanner can accurately obtain three-dimensional morphology of asphalt pavement texture, and this study proposed a texture section method to describe the 3D distribution of road surface texture at different depths. The macrotexture of the road surface gradually changes from sparse to dense starting from the shallow layer to the lowest layer. The actual asphalt pavement texture can be characterized by a simplified combination surface model of "cone + sphere + column".

(3) Based on the three-dimensional surface model of asphalt pavement texture reconstructed by the laser scanning method, the surface area distributions of macro and microtextures of different pavements are respectively calculated. Results show that the surface texture distribution of asphalt pavement has an obvious scale effect. The microtexture scale is the smallest, and the surface area of its surface texture is the largest, which is about 1.8–2.2 times of the sectional area. The surface area of the macrotexture is about 1.4 times larger than the sectional area but smaller than the surface area of micro convex body.

(4) The texture distribution density index can characterize the roughness of different pavements. The $S_{MA}$ index can represent the macroscopic structure level of different asphalt pavements to a certain extent, and the $S_{MI}$ index can well represent the friction level of different asphalt pavements. It can serve as a reference for multi-scale characterization of asphalt pavement textures.

## 6. Prospects

In this paper, a three-dimensional evaluation method is proposed for the macroscopic and microtexture features of different asphalt pavements, which can effectively characterize the texture of asphalt pavement. Moreover, the correlations between macrotexture and skid resistance as well as microtexture and skid resistance, respectively, are initially established. However, the wear of pavement texture causes the attenuation of skid resistance performance. On the one hand, it is affected by the secondary compaction of asphalt mixture; on the other hand, it is mainly affected by the contact and polishing of the tire on the top of the texture. Further research should expand more test samples (different pavement types, different degrees of wear, different minerals) and fully consider the effective contact and friction mechanism of the pendulum or tire, the finer microtextures scanned by higher-resolution equipment, and the difference in the hydrophilicity of different rock minerals.

**Author Contributions:** Conceptualization, B.C. and C.X.; methodology, B.C.; investigation, C.X. and W.L.; formal analysis, C.X. and X.Z.; writing—review and editing, B.C., C.X., W.L. and J.H. All authors have read and agreed to the published version of the manuscript.

**Funding:** The authors would like to acknowledge the financial support provided by the "China Postdoctoral Science Foundation" (2020M672639).

**Institutional Review Board Statement:** Not applicable.

**Informed Consent Statement:** Informed consent was obtained from all subjects involved in the study.

**Data Availability Statement:** The data presented in this study are available on request from the corresponding author.

**Conflicts of Interest:** The authors declare no conflict of interest.

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
