# Peer review of "Assessing Surface Texture Features of Asphalt Pavement Based on Three-Dimensional Laser Scanning Technology"

_buildings, doi:10.3390/buildings11120623_

Round 1
Reviewer 1 Report
In my opinion, the article has got a potential, although I would recommend to authors to answer my questions (to themselves). It would be also helpful to add some of these answers into the text.
Line 41-42 – It would be more appropriate to state the original source of such dividing. At the World Congress in Brussels in 1987, based on physical relationships between texture and friction/noise, etc., the World Road Association (PIARC) originally defined the ranges of micro-, macro- and megatexture.
Optimization of Surface Characteristics. Report to the XVIIIth World Road Congress 1987 in Brussels, Belgium, from the Technical Committee on Surface Characteristics, World Road Association (PIARC) (formerly the Permanent International Association of Road Congresses), Paris,1987
or the standard ISO 13473-2:2002.
Line 43 – „the macroscopic and microscopic texture“ - macro- and microtexture are more commonly used terms. I recommend using them.
Line 60 – 62 – I recommend rephrasing the sentence. 2D profile measurements can have a high resolution up to the microtexture level. Compare 2D and 3D methods and resolution separately.
Line 63 – 74 – I recommend rephrasing the whole paragraph. I think the point is escaping here. The goal is not to find the objective morphology information but to use the contactless measurement to predict the friction coefficient.
Line 103 – 104 – I recommend rephrasing the sentence. 3D measurements do not necessarily have to mean a capturing of microtexture.
Line 144 – I recommend completing the description of the parts of the equipment.
Line 172 – 174 – Was there performed any comparison between results obtained using different method interpolation?
Line 188 – Is the Figure 3(a) OK? Isn't it rotated 90 °?
Line 195 – 197 – How can be the length and width controlled at ±3 μm, or ±1.6 μm respectively when the maximum device resolution is 50 μm?
Line 214 – 224 – In terms of the substance of the article, Tables 1, 2, 3, 4 are redundant.
Line 231 – 234 and Figure 6 – The specimens were 300x300 mm. Figure 6 shows a scanned area of 180x180mm. How does it correspond with sampling interval 0.5 mm (line 247) and a number of profiles 180 (line 257)? It should be clearly stated what area was scanned.
Also, if the scanning was performed on new specimens with asphalt film on aggregates. Was there any problem with reflections?
Line 243 – 248 – With the maximum device resolution of 50 μm the minimum wavelength which can be objectively detected is at best 0,2 mm which represents only the upper (very small) part of the microtexture.
Line 253 – „the root mean square deviation of contour“ - instead of a "contour" a "profile" would be a better term.
Line 257 – 258 – „Each pavement has 180 contours, and the root mean square deviation of contour of four types of pavement is calculated respectively.“ I recommend rephrasing the sentence. It is not clear. 180 profiles? If the specimen is 300x300mm and the maximum resolution is 50 um then there should be 6000 profiles for evaluation. Besides, in figure 7, X-axis is denoted as "Contour number", with the range up to cca 260, please explain.
Line 297 – „90mmx90cm“ - it is probably a typo. Anyway, it corresponds better to sampling interval 0,5 mm (line 247) and number of profiles 180 (line 257). It should be unified a stated clearly in section 3.2
Line 297 – 299 – I recommend rephrasing the sentence. "...number of textures" is not an appropriate term in this meaning.
Line 307 – 308 – What values of radius of sphere model were taken into the calculation and how was it determined?
Line 312 – What values of the top angle of the cone were taken into the calculation and how was it determined?
Line 347 – Figure 12, The maximum value of the measured area approaches the value of 32,400, which corresponds to a scan size of 180x180, is my assumption correct? If so, this should also be stated in connection with the comment at line 233
Line 366 – 367 – Figure 13 shows a detail, but the microtexture is not visible. Is it caused by device resolution or asphalt film on the aggregate surface?
Line 376 – 378 – How was the microtexture surface area calculated at different depths? How was the microtexture separated from macrotexture, or how was macrotexture filtered out of the scanned surface, respectively? Primary surface, roughness surface, waviness surface.
Line 392 – 393 – There is no clear procedure for calculating the area of microtexture and macrotexture. Also, there were no stated parameters for the calculation of macrotexture sectional area.
Line 418 – Was the coefficient of friction measured on new specimens where the aggregates were covered with the asphalt film or was the film before the pendulum test removed? The info should be stated. If there is an asphalt film on aggregates then obtained values are (according to my experiences) too high for the wet surface.
Line 422 – Figure 15. The parameter of texture depth evaluated by the sand patch method is usually denoted as MTD. Unit of pendulum friction coefficient should be (-) instead of (BPN).
Line 430 – 432 – Considering the resolution of the measuring device, the asphalt film thickness, and resulting dispersion at GAC-16 and SMA-13 it is hardly possible to do any objective conclusion. This can be also affected by the way of determination microtexture and the filtering of primary surface, see comment above.
Line 443 – 444 – Was the correlation calculated from results obtained on all 16 specimens, or was taken into account only average values? If so, the conclusion can be hardly considered statistically significant and objective.
Line 447 – 449 – „The macroscopic texture distribution density can accurately reflect and evaluate the three-dimensional surface characteristics of the macroscopic texture of the pavement surface.“ - How so? According to figure 16, there is no difference between asphalt mixtures.
Line 459 – 464 – Same as the comment above.
Line 470 – 471 – „A single road surface contour is difficult to effectively characterize the roughness of asphalt pavement.“ - I recommend rephrasing the sentence. Besides, In my opinion, quite contrary, this was the best result for macrotexture characterization of different asphalt mixtures in this article.
In addition, I suggest adding a discussion on device resolution, asphalt film presence, primary surface filtering and microtexture determination, and statistical significance.
I also suggest discussing the authors’ results in view of the following references not yet mentioned in the article but which are highly relevant to it as recent research results (and as references, they need to be edited):
Zou, Y.; Yang, G.; Huang, W.; Lu, Y.; Qiu, Y.;Wang, K.C.P. Study of Pavement Micro- and Macro-Texture Evolution Due to Traffic Polishing Using 3D Areal Parameters. Materials 2021, 14, 5769. https://doi.org/10.3390/ma14195769
Kováč, M.; Brna, M.; Decký, M. Pavement Friction Prediction Using 3D Texture Parameters. Coatings 2021, 11, 1180. https://doi.org/10.3390/coatings11101180
and I also recommend inspiring in chapter 4.2 by title:
Minh Tan Do, Hassan Zahouani. Influence of the road-surface texture on the speed dependency of tire/road friction. 10th International Conference Metrology and Properties of Engineering Surfaces, Jul 2005, France. 11p., schémas, tabl., graphiques, ill. hal-00851275
Author Response
Open Review
(x) I would not like to sign my review report
( ) I would like to sign my review report
English language and style
(x) Extensive editing of English language and style required
( ) Moderate English changes required
( ) English language and style are fine/minor spell check required
( ) I don't feel qualified to judge about the English language and style
|
|
Yes |
Can be improved |
Must be improved |
Not applicable |
Does the introduction provide sufficient background and include all relevant references? |
( ) |
( ) |
(x) |
( ) |
Is the research design appropriate? |
( ) |
(x) |
( ) |
( ) |
Are the methods adequately described? |
( ) |
( ) |
(x) |
( ) |
Are the results clearly presented? |
( ) |
( ) |
(x) |
( ) |
Are the conclusions supported by the results? |
( ) |
( ) |
(x) |
( ) |
Comments and Suggestions for Authors
In my opinion, the article has got a potential, although I would recommend to authors to answer my questions (to themselves). It would be also helpful to add some of these answers into the text.
(1)Line 41-42 – It would be more appropriate to state the original source of such dividing. At the World Congress in Brussels in 1987, based on physical relationships between texture and friction/noise, etc., the World Road Association (PIARC) originally defined the ranges of micro-, macro- and megatexture.
Optimization of Surface Characteristics. Report to the XVIIIth World Road Congress 1987 in Brussels, Belgium, from the Technical Committee on Surface Characteristics, World Road Association (PIARC) (formerly the Permanent International Association of Road Congresses), Paris,1987
or the standard ISO 13473-2:2002.
Response: According to comments, the original source is adopted for the classification of pavement texture, and the reference information is updated.
(2)Line 43 – „the macroscopic and microscopic texture“ - macro- and microtexture are more commonly used terms. I recommend using them.
Response: The technical terminology in the manuscript has been updated according to the comments.
(3)Line 60 – 62 – I recommend rephrasing the sentence. 2D profile measurements can have a high resolution up to the microtexture level. Compare 2D and 3D methods and resolution separately.
Response: This sentence has been rewritten as suggested.
(4)Line 63 – 74 – I recommend rephrasing the whole paragraph. I think the point is escaping here. The goal is not to find the objective morphology information but to use the contactless measurement to predict the friction coefficient.
Response: This sentence has been rewritten as suggested.
(5)Line 103 – 104 – I recommend rephrasing the sentence. 3D measurements do not necessarily have to mean a capturing of microtexture.
Response: This sentence has been rephrased as suggested.
(6)Line 144 – I recommend completing the description of the parts of the equipment.
Response: According to the comments, the three-dimensional data display system and data analysis system have been supplemented. For details, please refer to the manuscript Line 142-151.
(7)Line 172 – 174 – Was there performed any comparison between results obtained using different method interpolation?
Response: Yes, I tried to use different interpolation methods for comparison in the early stage. From the image, due to the lack of data points, and at the bottom of the specimen, the interpolated data has very little effect on the subsequent calculation results. In addition, according to the principles and calculations of different interpolation methods, the use of linear interpolation can significantly improve the efficiency of data analysis.
(8)Line 188 – Is the Figure 3(a) OK? Isn't it rotated 90 °?
Response: Thanks for the comments. We have revised this figure.
(9)Line 195 – 197 – How can be the length and width controlled at ±3 μm, or ±1.6 μm respectively when the maximum device resolution is 50 μm?
Response: This is because the processing accuracy of the calibration block needs to be higher than the scanning accuracy of the equipment. This processing accuracy can be easily achieved in a professional manufacture in China according to the Chinese standard GB / T 6093-2001.
(10)Line 214 – 224 – In terms of the substance of the article, Tables 1, 2, 3, 4 are redundant.
Response: Thanks for the comments. Since this study mainly conducted research on different asphalt mixture samples formed indoors, to maintain the integrity of the test information of the paper, the author prefers to retain relevant test material information.
(11)Line 231 – 234 and Figure 6 – The specimens were 300x300 mm. Figure 6 shows a scanned area of 180x180mm. How does it correspond with sampling interval 0.5 mm (line 247) and a number of profiles 180 (line 257)? It should be clearly stated what area was scanned.
Also, if the scanning was performed on new specimens with asphalt film on aggregates. Was there any problem with reflections?
Response: (1) The equipment scans the 300*300mm test piece, and the obtained point cloud data is 6000*6000 rows (36 million data points). In the post-processing process, it is difficult for ordinary computers to efficiently process, or even to calculate related indicators. . After the tire ground impression test (CBM) of the previous research group, the tire impression width is generally within 175mm and the length is about 160~180cm under standard conditions; in addition, when the pavement structure depth and friction coefficient are tested by the sand paving method, the size of the test area is average It is also in the range of 120~200mm; at the same time, considering the calculation performance of ordinary office computers, the 180*180mm range in the middle of the rut plate specimen is selected for analysis and calculation. On the one hand, it can also establish a clearer correlation with the friction coefficient and structural depth indicators. Relevant information has been supplemented in chapter 4 of the manuscript.
(2) The freshly formed asphalt rut plate test piece, the asphalt film is relatively fresh, it is prone to specular reflection, forming noise in the scanned image, so the test piece is generally placed in the open for 2 days before scanning. In addition, noise can also be eliminated by using image processing methods such as noise reduction and filtering.
(12)Line 243 – 248 – With the maximum device resolution of 50 μm the minimum wavelength which can be objectively detected is at best 0,2 mm which represents only the upper (very small) part of the microtexture.
Response: From the XY plane, only a small part of the micro-texture can be identified. This is the limitation of the current 3D scanning equipment. However, from the Z direction, the scanning accuracy of the laser can still detect textures of 0.01mm. Therefore, we also believe that the device can reflect the topographical characteristics of some micro-textures to a certain extent.
(13)Line 253 – „the root mean square deviation of contour“ - instead of a "contour" a "profile" would be a better term.
Response: As suggested, the description of the outline in the manuscript has been revised.
(14)Line 257 – 258 – „Each pavement has 180 contours, and the root mean square deviation of contour of four types of pavement is calculated respectively.“ I recommend rephrasing the sentence. It is not clear. 180 profiles? If the specimen is 300x300mm and the maximum resolution is 50 um then there should be 6000 profiles for evaluation. Besides, in figure 7, X-axis is denoted as "Contour number", with the range up to cca 260, please explain.
Response: About 260 profiles on the surface of asphalt mixture specimens were extracted with 1mm spacing (40mm length at both ends was deleted considering the segregation of the specimen edges), and the root mean square deviation of profile of four types of pavement is calculated respectively.
The supplementary information can be found in section 4.1.
(15)Line 297 – „90mmx90cm“ - it is probably a typo. Anyway, it corresponds better to sampling interval 0,5 mm (line 247) and number of profiles 180 (line 257). It should be unified a stated clearly in section 3.2
Response: More information has been supplied in section 4.2.
(16)Line 297 – 299 – I recommend rephrasing the sentence. "...number of textures" is not an appropriate term in this meaning.
Response: This sentence has been rephrased according to the comments.
(17)Line 307 – 308 – What values of radius of sphere model were taken into the calculation and how was it determined?
Response: This paper mainly constructs the relationship between the cross-sectional area change rule of the spherical model and the cross-sectional depth. Therefore, according to the Pythagorean theorem, the radii of different circular cross-sections are converted into mathematical expressions of h. In addition, according to the exposed state of the road aggregate, the simplified sphere model only considers the upper hemisphere.
(18)Line 312 – What values of the top angle of the cone were taken into the calculation and how was it determined?
Response: For the simplified cone model apex angle value in this paper, since it does not change with the change of the cone section depth, it is regarded as a fixed coefficient in Equation (6), and figure 10 is drawn accordingly.
(19)Line 347 – Figure 12, The maximum value of the measured area approaches the value of 32,400, which corresponds to a scan size of 180x180, is my assumption correct? If so, this should also be stated in connection with the comment at line 233
Response: Yes, the maximum value of the measurement area is the plane area, which is 32400mm2. Supplementary explanations have been added in section 4 based on the comments.
(20)Line 366 – 367 – Figure 13 shows a detail, but the microtexture is not visible. Is it caused by device resolution or asphalt film on the aggregate surface?
Response: Because the test piece used in this paper is the asphalt rutting slab, the coverage of the asphalt film makes it difficult to measure the microscopic texture of the aggregate.
(21)Line 376 – 378 – How was the microtexture surface area calculated at different depths? How was the microtexture separated from macrotexture, or how was macrotexture filtered out of the scanned surface, respectively? Primary surface, roughness surface, waviness surface.
Response: As it is difficult to define and count the particle adhesion of asperities, the current evaluation of the microstructure of the structure surface has not yet reached a unified conclusion. This paper distinguishes the micro and macro textures by setting different scanning resolutions. The road texture data collected at a resolution of> 0.5mm is regarded as the macro structure; the road texture data collected at a resolution of 0.05mm (the highest resolution) is regarded as Micro + macro structure, as micro texture data calculation and analysis.
(22)Line 392 – 393 – There is no clear procedure for calculating the area of microtexture and macrotexture. Also, there were no stated parameters for the calculation of macrotexture sectional area.
Response: This article mainly uses the projection method in section 3.3, and writes a program in MATLAB software to calculate the micro-texture surface area, macro-texture surface area, and cross-sectional area at different depths. For discrete spatial point cloud data, given point spacing parameters, different area results can be calculated by iterative row by row. According to the comments, it is added in section 3.3.
(23)Line 418 – Was the coefficient of friction measured on new specimens where the aggregates were covered with the asphalt film or was the film before the pendulum test removed? The info should be stated. If there is an asphalt film on aggregates then obtained values are (according to my experiences) too high for the wet surface.
Response: The coefficient of friction was tested on a new specimen covered with an asphalt film, and the test conditions were added to the manuscript based on comments. The friction coefficient level of this new specimen is indeed a bit high. We have therefore carried out a lot of repeated tests to eliminate human error, but they all showed relatively stable results, which may be related to the used aggregate, asphalt, mixture gradation, and even test equipment. In this study, the difference in friction coefficient of different specimens is still a meaningful finding.
(24)Line 422 – Figure 15. The parameter of texture depth evaluated by the sand patch method is usually denoted as MTD. Unit of pendulum friction coefficient should be (-) instead of (BPN).
Response: Figure 15 has been revised as suggested.
(25)Line 430 – 432 – Considering the resolution of the measuring device, the asphalt film thickness, and resulting dispersion at GAC-16 and SMA-13 it is hardly possible to do any objective conclusion. This can be also affected by the way of determination microtexture and the filtering of primary surface, see comment above.
Response: According to our parallel test results, the three-dimensional morphological difference of the surface structure of different types of asphalt mixture specimens is more significant. The thickness of the asphalt film initially covers part of the micro texture, but the thickness and gradation of the asphalt film will also have different effects. We will continue to carry out research on the attenuation behavior of anti-skid performance of different asphalt pavements under the action of vehicles, with a view to revealing the changing laws of the three-dimensional texture of asphalt pavements under different usage conditions.
(26)Line 443 – 444 – Was the correlation calculated from results obtained on all 16 specimens, or was taken into account only average values? If so, the conclusion can be hardly considered statistically significant and objective.
Response: The correlation of the data in Table 7 is calculated by the average of the results of different roads. In order to avoid misunderstandings, the author has made a supplementary explanation in section 4.4. In our research, we found that in the same pavement type samples, SMI and BPN have relatively large fluctuations. The preliminary analysis is mainly related to the segregation of the specimen forming process and the precise positioning of the test position. Therefore, we chose different asphalts. The pavement type is used as a sample for correlation analysis. This part of the content is also discussed in chapter 4.4 of the manuscript.
(27)Line 447 – 449 – „The macroscopic texture distribution density can accurately reflect and evaluate the three-dimensional surface characteristics of the macroscopic texture of the pavement surface.“ - How so? According to figure 16, there is no difference between asphalt mixtures.
Response: According to the calculation results, the macro texture distribution density difference of different asphalt specimens is relatively small, and the calculated value is between 1.387~1.413, which is mainly related to the resolution of the set equipment. From the per-spective of the correlation model, the index has a good correlation with the structural depth index, which can reflect the macro-texture state of the road surface to a certain extent.
(28)Line 459 – 464 – Same as the comment above.
Response: In contrast, the micro-texture density index difference between different asphalt pavements is obvious, and the correlation coefficient of the correlation model of different pavement parameters is better. It can be considered that this index can better reflect the friction of different pavements. level.
(29)Line 470 – 471 – „A single road surface contour is difficult to effectively characterize the roughness of asphalt pavement.“ - I recommend rephrasing the sentence. Besides, In my opinion, quite contrary, this was the best result for macrotexture characterization of different asphalt mixtures in this article.
Response: Based on comments, the conclusion and summary have been revised. The pavement profile index of the full section can reflect the macroscopic structure of the road surface, but one or several contours obviously have the shortcoming of insufficient representation. It is necessary to establish a three-dimensional evaluation index of the road surface structure.
(30)In addition, I suggest adding a discussion on device resolution, asphalt film presence, primary surface filtering and microtexture determination, and statistical significance.
I also suggest discussing the authors’ results in view of the following references not yet mentioned in the article but which are highly relevant to it as recent research results (and as references, they need to be edited):
Response: As suggested, further discussion has been added in the section 4.4. In addition, the following references were cited in the revised manuscript.
Zou, Y.; Yang, G.; Huang, W.; Lu, Y.; Qiu, Y.;Wang, K.C.P. Study of Pavement Micro- and Macro-Texture Evolution Due to Traffic Polishing Using 3D Areal Parameters. Materials 2021, 14, 5769. https://doi.org/10.3390/ma14195769
Kováč, M.; Brna, M.; Decký, M. Pavement Friction Prediction Using 3D Texture Parameters. Coatings 2021, 11, 1180. https://doi.org/10.3390/coatings11101180
and I also recommend inspiring in chapter 4.2 by title:
Minh Tan Do, Hassan Zahouani. Influence of the road-surface texture on the speed dependency of tire/road friction. 10th International Conference Metrology and Properties of Engineering Surfaces, Jul 2005, France. 11p., schémas, tabl., graphiques, ill. hal-00851275
Response: As suggested, the title of chapter 4.2 has been revised to inspire the simplified characterization model of pavement texture.
Submission Date
07 November 2021
Date of this review
18 Nov 2021 16:21:37

Reviewer 2 Report
This is interesting and informative research. This manuscript is fairly well written, but still needs are review and editing by a Technical English editor. There are unusual word choices for example, Page 1, row 31, Page 18, row 463, Page 19, row 485 and row 491: the word “richness” should be replaced with “the degree of”. There are incomplete sentences such as Page 2, rows 88-89: This is not a complete sentence or after “small” should a word be added or the remainder of the sentence be re-written?
Another example is Page 2, rows 59-60: the portion of the sentence …greatly affected by human factors resulting in large dispersion and low accuracy. This should be written in term of variability, precision, bias and accuracy which terms used in ASTM standard test method and the precision statements from the test methods should be used to communicate the actual precision and bias. No other examples will be listed, the point is the manuscript needs to edited by technical English editor.
Page 5, rows 177-179: …of the equipment… what does this mean?
In general figures and table should be introduced in the manuscript before being referenced. For example Page 11, Figure 7 and Table 6 are placed before being described. At the end of row 258 a sentence should be added stating figure 7 illustrates…and table 6 is a summary of…
Page 16, figure 14. Use a common x-axis scale for all plots in figure 14 so difference are more apparent.
Page 19, row 496: the word “roughness” should be replaced with “texture”
Author Response
Open Review
(x) I would not like to sign my review report
( ) I would like to sign my review report
English language and style
(x) Extensive editing of English language and style required
( ) Moderate English changes required
( ) English language and style are fine/minor spell check required
( ) I don't feel qualified to judge about the English language and style
|
|
Yes |
Can be improved |
Must be improved |
Not applicable |
Does the introduction provide sufficient background and include all relevant references? |
(x) |
( ) |
( ) |
( ) |
Is the research design appropriate? |
(x) |
( ) |
( ) |
( ) |
Are the methods adequately described? |
(x) |
( ) |
( ) |
( ) |
Are the results clearly presented? |
( ) |
( ) |
(x) |
( ) |
Are the conclusions supported by the results? |
( ) |
(x) |
( ) |
( ) |
Comments and Suggestions for Authors
(1)This is interesting and informative research. This manuscript is fairly well written, but still needs are review and editing by a Technical English editor. There are unusual word choices for example, Page 1, row 31, Page 18, row 463, Page 19, row 485 and row 491: the word “richness” should be replaced with “the degree of”. There are incomplete sentences such as Page 2, rows 88-89: This is not a complete sentence or after “small” should a word be added or the remainder of the sentence be re-written?
Response: The grammatical errors of the manuscript have been corrected by a language editor.
(2)Another example is Page 2, rows 59-60: the portion of the sentence …greatly affected by human factors resulting in large dispersion and low accuracy. This should be written in term of variability, precision, bias and accuracy which terms used in ASTM standard test method and the precision statements from the test methods should be used to communicate the actual precision and bias. No other examples will be listed, the point is the manuscript needs to edited by technical English editor.
Response: The language of this manuscript has been polished by an English-speaking editor。
(3)Page 5, rows 177-179: …of the equipment… what does this mean?
response: Sorry for the unclear statement. This sentence has been revised as ‘In the actual scanning process of the equipment, the scanning plane of the laser equipment is not completely parallel to the road surface due to the unevenness of the pavement’.
(4)In general figures and table should be introduced in the manuscript before being referenced. For example Page 11, Figure 7 and Table 6 are placed before being described. At the end of row 258 a sentence should be added stating figure 7 illustrates…and table 6 is a summary of…
Response” Supplementary information have been added as suggested。
(5)Page 16, figure 14. Use a common x-axis scale for all plots in figure 14 so difference are more apparent.
Response Figure 14 has been revised as suggested.
(6)Page 19, row 496: the word “roughness” should be replaced with “texture”
Response: ‘roughness’ has been replaced by ‘texture’ as suggested.
Submission Date
07 November 2021
Date of this review
13 Nov 2021 17:48:53
